# Genome-Wide Analysis of Homologous E6-AP Carboxyl-Terminal E3 Ubiquitin Ligase Gene Family in *Populus trichocarpa*

**Yanrui Fu [1], Hui Li [1], Yaqi Li [1], Haoqin Zhao [1], Da Yang [1], Aihua Chen [2,* and Jingli Yang [1,*]**

1   State Key Laboratory of Tree Genetics and Breeding, Northeast Forestry University, Harbin 150040, China;
    fuyanrui@nefu.edu.cn (Y.F.); lihuihuihui0916@163.com (H.L.); lyq2896801078@163.com (Y.L.);
    zhqftb@163.com (H.Z.); yd15192971926@126.com (D.Y.)
2   Key Laboratory of Non-Wood Forest Product Research and Development, Mudanjiang Branch of
    Heilongjiang Academy of Forestry, 16 Diming Street, Mudanjiang 157011, China
*   Correspondence: aihua_0117@nefu.edu.cn (A.C.); yifan85831647@nefu.edu.cn (J.Y.)

**Abstract:** Proteins containing the homologous E6-AP carboxyl-terminal (HECT) domain are a class of E3 ubiquitin ligases involved in the ubiquitin–proteasome pathway, which plays an irreplaceable role in plant growth, development, and stress resistance. However, a phylogenetic analysis and expression profile of the *HECT* gene (*PtrHECT*) in the model plant *Populus trichocarpa* (Torr. & Gray) have not been reported. In this study, we identified 13 *PtrHECT* genes using genome-wide analysis, and then these were divided into four groups. The protein interaction networks showed that the PtrHECT protein may interact with PTR6 and participate in ABA signal regulation. Abiotic stress is the main environmental factor limiting plant growth and development. The qRT-PCR results showed that *PtrHECT1*, *4*, *7*, *8*, and *9* were significantly up-regulated in leaves at each time point under drought stress, and most *PtrHECT* genes responded to both drought and high salt stress, consistent with their promoter sequence analysis, revealing the presence of an important number of phytohormone-responsive and stress-related *cis*-regulatory elements. This study provides useful information for further analysis of the functions of the *HECT* gene family in *P. trichocarpa*.

**Keywords:** expression analysis; gene identification; *HECT* genes family; *Populus trichocarpa*





## 1. Introduction

The ubiquitin-26S proteasome system (UPS) is one of the most important protein degradation systems in eukaryotes. About 80% of proteins are degraded by the ubiquitin–proteasome pathway [1–4]. Ubiquitination specifically regulates protein expressions at the post-translational level to achieve the precise regulation of plant growth and development, metabolism, and response to biotic and abiotic stresses [5–7]. Ubiquitination is a three-enzyme cascade, which generally requires three enzymes: ubiquitin-activating enzyme (E1), ubiquitin-conjugating enzyme (E2), and ubiquitin ligase (E3) [8–10].

According to the subunit composition, E3 can be divided into two categories. Those with a single subunit structure are classified into one category, including RING (a really interesting new gene), U-box, and HECT (homology to the E6-AP C-terminus). Another type with a multi-subunit structure includes CRLs (cullin-RING), APC, and SCF [4,11–13]. The HECT-type E3 protein has a characteristic HECT domain composed of two N/C lobes, which consist of about 350 amino acids at the C-terminus [14,15]. Among these, the cysteine residues at 32–36 amino acids of the C-terminal lobe are highly conserved, and the N-terminal lobe contains the E2-binding site [16,17].

In this process, ubiquitin is attached to E2 to form a complex, which is recruited by E3, and then ubiquitin is connected to the cysteine site at the C-terminus of E3 by forming a thioester and transferred to the lysine residue of the target protein, which is degraded into

small molecular peptides in the 26S proteasome [17,18]. The N-terminus of the HECT-type E3 protein specifically recognizes the degradation signal on the various substrates, and the C-terminus indirectly or directly catalyzes the transfer of ubiquitin, thereby strictly controlling the efficiency of the ubiquitination reaction [16,19]. Based on the different N-terminal domains that recognize and bind to varied substrates, HECT-type E3 can be divided into different subfamilies.

Studies have shown that the *HECT* gene family plays a key role in multiple developmental stages of plants (such as photomorphogenesis, cell cycle, ovule development, senescence, defense, environmental response, and hormone signal response) [12,18]. At present, a total of seven HECT-type E3 ubiquitin ligases (UPL1–UPL7) have been identified and fully studied in *Arabidopsis thaliana* [20,21]. Among them, UPL3 is involved in the regulation of plant epidermal trichome development by regulating the degradation rate of related transcription factors participating in leaf morphogenesis [22,23]. UPL3 regulates the nuclear replication cycle and seed size as well [24]. UPL4 is involved in physiological activities such as active oxygen balance and iron metabolism. UPL5 regulates the ubiquitination of WRKY53 (a plant-specific zinc finger transcription regulatory factor), thereby regulating the aging process [25]. Studies have shown that UPL1, UPL3, and UPL5 play a role in plant immunity and salicylic acid (SA) responses [26]. In rice (*Oryza sativa*), seven *HECT* genes have been identified [25]. In soybean (*Glycine max*), 19 *HECT* genes have been found, and 15 genes were shown to be differentially expressed in tissues and usually highly expressed in flowers and roots [27]. In addition, *HECT* genes have been identified and analyzed in apple (*Malus domestica*), tomato (*Solanum lycopersicum*), wheat (*Triticum aestivum*), maize (*Zea mays*), and *Brassica rapa* [25,28–30]. The results of multiple studies have revealed that the *HECT* gene family may play an important role in plant adaptation to adverse environmental stresses. However, similar results have not been seen in studies focused on poplar plants, represented by *Populus trichocarpa*.

*P. trichocarpa* is an important economic and scientific tree species with characteristics of fast growth and easy propagation. Since the genome data of *P. trichocarpa* were first published in 2006, research has gradually deepened [31–33]. In this study, we obtained the *HECT* gene sequences of *P. trichocarpa* and analyzed the evolutionary relationship, gene structures, conserved motifs, promoter *cis*-elements, homology models, protein interaction network, and expression profiles. Then, the relative expression levels of *PtrHECT* genes in roots and leaves under drought stress and salinity treatments were analyzed. The excavation of *PtrHECT* lays a theoretical foundation for exploring the role of the *HECT* gene family in growth and stress resistance, and this will hopefully help to improve plant resistance by molecular means in the future.

## 2. Materials and Methods

### 2.1. Identification of HECT Genes in P. trichocarpa

The amino acid sequences of 7 AtHECT proteins (AT1G55860, AT1G70320, AT4G38600, AT5G02880, AT4G12570, AT3G17205, and AT3G53090) were obtained as reference sequences from TAIR (https://www.arabidopsis.org/, accessed on 15 October 2023) [20]. Online alignment was performed using the Phytozome database of *P. trichocarpa* (version 13; https://phytozome-next.jgi.doe.gov/, accessed on 15 October 2023), and the following parameters were set: target type, proteome; program, BLASTP; expect (E) threshold = −1 [34]. At the same time, the hidden Markov model (PF00632) of the HECT domain was downloaded by the Pfam database (http://pfam.xfam.org/, accessed on 17 October 2023) [35]. The HMM search in the TBtools software (v1.064) was used to screen the HECT protein sequences from the total proteome database of *P. trichocarpa* [36,37]. Combining the above two methods, the repetitive sequences were detected and deleted. All candidate genes were confirmed through the Pfam and SMART databases (https://smart.embl.de/, accessed on 20 October 2023) to ensure the existence of a conserved HECT domain [38]. The gene sequences and coding amino acid sequences of *PtrHECT* were downloaded from Phytozome. Subcellular localization was predicted using WoLF PSORT (http://www.genscript.com/

psort/wolf_psort.html, accessed on 23 October 2023) [39]. The isoelectric point, amino acid length, molecular weight, and hydrophobicity of each PtrHECT protein were predicted using the online ProtParam tool of Expasy (https://web.expasy.org/protparam/, accessed on 23 October 2023) [40].

## 2.2. Chromosomal Location

The chromosome location information for 13 *PtrHECT* was obtained from the Phytozome database. The MapGene2 Chromosome online website (http://mg2c.iask.in/mg2c_v2.0/, accessed on 23 October 2023) was used for mapping.

## 2.3. Phylogenetic Analysis and Sequence Analysis

Using *PtrHECT* genes as the query sequences for standard Protein BLAST in the NCBI database (https://www.ncbi.nlm.nih.gov/, accessed on 8 November 2023), we obtained the amino acid sequences of HECT proteins from *P. trichocarpa*, *A. thaliana*, soybean (*G. max*), apple (*M. domestica*), rice (*O. sativa*), maize (*Z. mays*), and wheat (*T. aestivum*). These HECT protein sequences were subjected to multiple sequence alignment analyses using ClustalW of MEGA software (v11.0.10), and the conserved residues of the HECT domain were identified [41]. The neighbor-joining method was used to draw the phylogenetic tree. The test method used for phylogeny was the bootstrap method, and the number of bootstrap replications was set to 1000. The evolutionary distances were computed using the p-distance method. The remaining parameters were the default values. Conserved domain analysis of the PtrHECT protein was performed using the SMART database and plotted using IBS software (v1.0.3).

The genetic sequences and CDS of the *PtrHECT* were downloaded from the Phytozome online database, and the distributions of the exons and introns were visualized using the Gene Structure Display Server (http://gsds.gao-lab.org/, accessed on 8 November 2023) [42]. The MEME online tool (Multiple EM for Motif Elicitation, http://meme-suite.org/index.html, accessed on 10 November 2023) was used to analyze the conserved motifs of the full-length PtrHECT proteins [43]. The maximum number of motifs retrieved was 15, and the remaining parameters were the default values [26].

## 2.4. Promoter cis-Element Analysis

The promoter sequences of the *PtrHECT* were downloaded using the Phytozome online database. In general, a 2000 bp sequence upstream of the transcription start site is used as the promoter of the gene. The search and localization of cis-acting elements in each promoter were conducted through the online database PlantCARE, and 12 *cis*-acting elements related to plant hormones and abiotic stresses were selected for visualization in IBS software (v1.0.3) [44].

## 2.5. Protein Interaction Network Analysis

The protein interaction networks were obtained through the STRING (https://string-db.org/cgi/input.pl, accessed on 12 November 2023) database [45]. For drawing, we used Adobe Illustrator CS6 (v16.0.0).

## 2.6. Homology Model of HECT Genes

The protein sequences were submitted to the SWISS-MODEL Server (http://swissmodel.expasy.org/, accessed on 15 November 2023) to develop a model with sufficient query sequence coverage and sequence identity. The most reliable 3D structure was selected based on the Global Model Quality Estimation (GMQE) and Qualitative Model Energy Analysis (QMEAN) values [46]. After the model was predicted, we used SAVES (https://saves.mbi.ucla.edu, accessed on 16 November 2023) to evaluate and provide three software evaluation results at one time. If two of the results show approval, the model is considered usable. For instance, PROCHECK was used to check the quality of the *HECT* genes' generated modeled 3D structure via SWISS-MODEL. Generally, if the amino acid residues in the

acceptable region are greater than 90%, the protein structure can be considered reasonable. Verify3D was used to analyze the compatibility of the atomic model (3D) with its own amino acid sequence (1D), which was determined by assigning structure classes according to its location and environment and comparing the results with good structures. If more than 80% of the residues have a 3D/1D value greater than 0.2, the quality of the model is considered acceptable, but the parts below 0.2 require further correction. Also, ERRAT was used to collect data statistics for non-bonding interactions between different atomic types and plot the position relationship between the error function value and the residue sliding window, which was derived by comparing it with statistics for highly refined structures.

### 2.7. Tissue-Specific Expression Analyses

The RNA-Seq data (GSE81077) of *P. trichocarpa* were obtained using the GEO DataSets tool of NCBI. The experimental materials were the tissues (root, stem, leaf, xylem, phloem, fiber, and vessel) of 6-month-old *P. trichocarpa* [47]. The heat map was drawn using TBtools.

### 2.8. Plant Materials, Drought Stress, and Salt Treatment

*P. trichocarpa* (genotype Nisqually-1) plants were cultured until they reached three weeks old in vitro as the experimental material, and the basic medium was woody plant medium (WPM, a plant culture medium used for root propagation and elongation containing $CaCl_2$ and vitamins) [48,49]. We added 6.5 g/L of agar and 30 g/L of sucrose when configuring. The culture conditions were 25 °C and a 16 h photoperiod. Using untreated plants as the controls, 7% polyethylene glycol 6000 was added to one group to simulate drought, while 200 mM NaCl was added to the other group to simulate high salinity. Samples were taken from leaves and roots at 0, 3, 6, 12, and 24 h after stress, while unstressed samples were taken as the controls. The samples were then frozen in liquid nitrogen and placed in an ultra-low-temperature refrigerator.

### 2.9. RNA Extraction and Real-Time Quantitative PCR (qRT-PCR)

RNA reverse transcription TransScript®One-Step gDNA Removal and cDNA Synthesis SuperMix and fluorescence quantitative detection kits were purchased from TransGen Biotech and Vazyme.

The design of specific primers for qRT-PCR using the online NCBI Primer Blast tool and the *PtrActin* gene (GenBank ID: XM_002298674) were used as the internal reference genes. The information on all specific primers is listed in Supplementary Table S1.

### 2.10. Statistical Analysis

Statistical software SPSS (v27, IBM, Chicago, IL, USA) was used to analyze the differences in the relative expression, and significance tests of these differences ($p = 0.05$, $p = 0.01$) were carried out.

## 3. Results

### 3.1. Identification of HECT Genes in P. trichocarpa

In this study, a total of 13 *HECT* family members were identified in the genome of *P. trichocarpa*, which were named *PtrHECT1* ~ *PtrHECT13* according to their position on the chromosomes. The length of the PtrHECT proteins ranged from 782 aa (PtrHECT11) to 3756 aa (PtrHECT1) (Table 1). The molecular weight ranged from 88.83 kD to 411.29 kD. The isoelectric point ranged from 4.95 to 7.88. Except for PtrHECT13, all other family members were alkaline proteins. The GARVY values ranged from −0.315 to −0.105, indicating that the whole family had the characteristics of low hydrophilicity. PtrHECT1, PtrHECT3, PtrHECT4, PtrHECT5, PtrHECT7, PtrHECT9, PtrHECT10, and PtrHECT11 are localized in the nucleus. PtrHECT2 is located in the plasma membrane; PtrHECT13 is located in chloroplasts; PtrHECT6 and PtrHECT8 are located in the cytoplasm; and PtrHECT12 is located in the endoplasmic reticulum.

**Table 1.** Characteristic analysis of *PtrHECT* family genes.

| Gene Name | Gene ID | Protein Length (a.a.) | Molecular Weight (Da) | Theoretical pI | GRAVY | The Predicted Location of PtrHECT Proteins |
|---|---|---|---|---|---|---|
| *PtrHECT1* | Potri.001G368600 | 3756 | 411,293.51 | 4.95 | −0.213 | Nuclear |
| *PtrHECT2* | Potri.002G110500 | 3667 | 405,141.71 | 5.1 | −0.281 | Plas |
| *PtrHECT3* | Potri.004G174700 | 1877 | 201,786.1 | 5.97 | −0.308 | Nuclear |
| *PtrHECT4* | Potri.006G011700 | 840 | 953,69.61 | 6.34 | −0.129 | Nuclear |
| *PtrHECT5* | Potri.006G132000 | 1074 | 119,539.13 | 5.15 | −0.164 | Nuclear |
| *PtrHECT6* | Potri.008G101300 | 1033 | 117,999.93 | 6.75 | −0.161 | Cytoplasmic |
| *PtrHECT7* | Potri.009G134300 | 1895 | 203,638.89 | 5.62 | −0.315 | Nuclear |
| *PtrHECT8* | Potri.010G150000 | 1032 | 118,051.54 | 6.41 | −0.178 | Cytoplasmic |
| *PtrHECT9* | Potri.011G094100 | 3749 | 411,209.5 | 4.96 | −0.214 | Nuclear |
| *PtrHECT10* | Potri.016G012900 | 853 | 97,311.88 | 6.24 | −0.18 | Nuclear |
| *PtrHECT11* | Potri.016G059800 | 782 | 88,833.73 | 5.95 | −0.192 | Nuclear |
| *PtrHECT12* | Potri.016G085200 | 1512 | 169,298.31 | 5.31 | −0.11 | E.R._plas |
| *PtrHECT13* | Potri.016G096500 | 1173 | 132,025.06 | 7.88 | −0.105 | Chloroplast |

### 3.2. Chromosomal Locations

Using the Phytozome database, it was confirmed that thirteen *PtrHECT* genes were unevenly distributed on 9 out of 18 chromosomes. Four *PtrHECT* genes (*PtrHECT10*, *11*, *12*, and *13*) were located on chromosome 16. Two *PtrHECT* genes (*PtrHECT4* and *5*) were located on chromosome 6. Additionally, chromosomes 1, 2, 4, 8, 9, 10, and 11 each contained one *PtrHECT* gene (Figure 1).

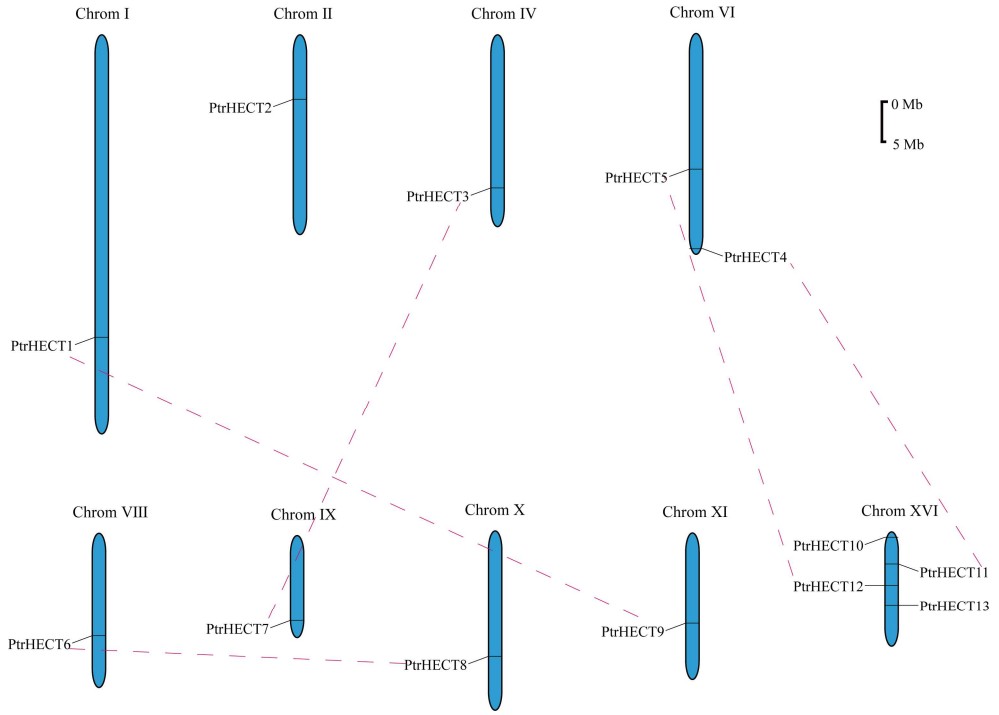

**Figure 1.** The chromosome locations and segmental paralogous pairs of *PtrHECT* gene family members.

### 3.3. Sequence Alignment, Phylogenetic Analysis, and Domain Assessment

Analysis with PROSITE and multiple sequence alignment confirmed the appearance of nine highly conserved cysteine residues in the HECT domain of multiple species, in addition to conserved lysine residues (Supplementary Figure S1). An unrooted phylogenetic tree was constructed using the amino acid sequences of 13 HECT proteins from *P. trichocarpa*, 7

from *A. thaliana*, 13 from apple, 7 from rice, 19 from soybean, 12 from maize, and 25 from wheat (Figure 2). The results showed that *PtrHECTs* were divided into four subfamilies. Subfamily I had the largest number of members (4). *PtrHECT3*, *PtrHECT5*, *PtrHECT7*, and *PtrHECT12* belong to subfamily I; *PtrHECT4*, *PtrHECT10*, and *PtrHECT11* belong to subfamily II; *PtrHECT6*, *PtrHECT8*, and *PtrHECT13* belong to subfamily III; and *PtrHECT1*, *PtrHECT2*, and *PtrHECT9* belong to subfamily IV. PtrHECT proteins are mostly clustered with the HECT proteins of apple, indicating that the HECT proteins of *P. trichocarpa* are more closely related to the homologous proteins of apple than to the homologous proteins of soybean, *A. thaliana,* and other species.

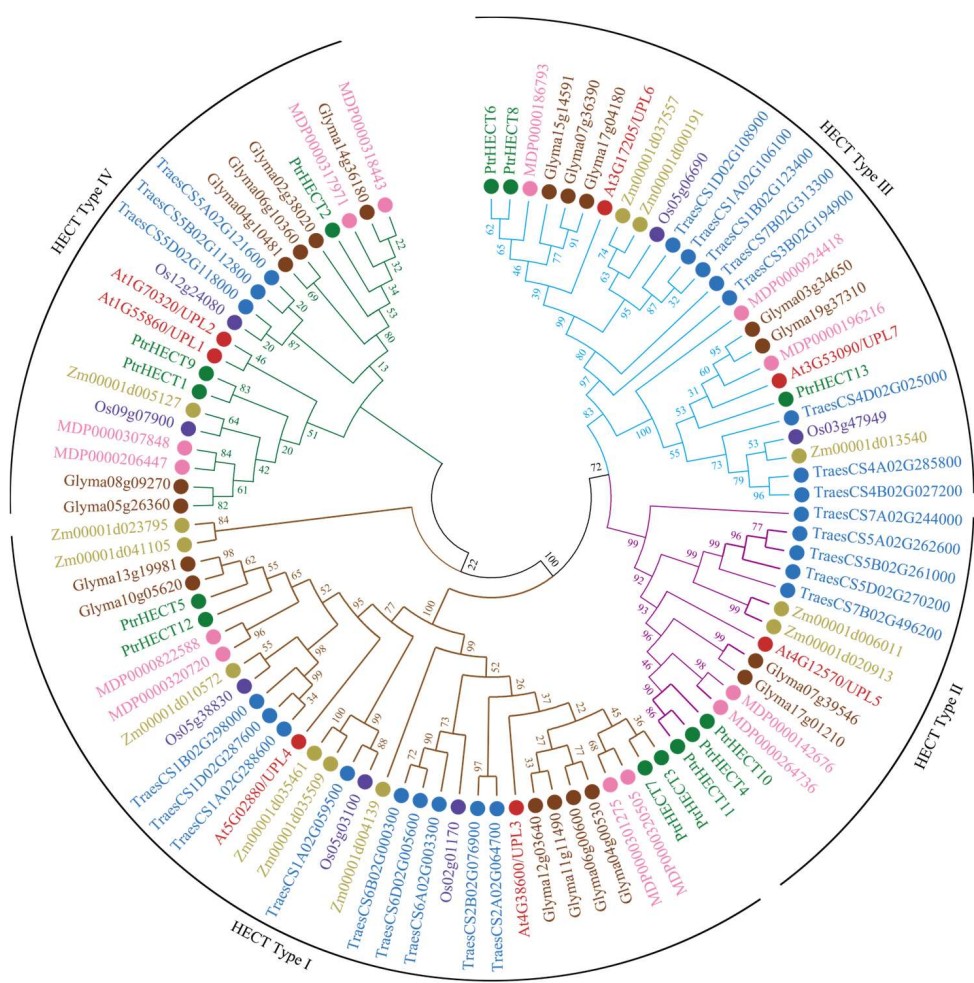

**Figure 2.** Phylogenetic tree of the *HECT* gene family in *Populus trichocarpa*, *Arabidopsis thaliana*, *Malus domestica*, *Glycine max*, *Oryza sativa*, *Zea mays* and *Triticum aestivum*.

In all cases, the HECT domain was located at the C-terminal, and there were differences in the N-terminal between subfamilies (Figure 3). The HECT proteins were divided into four classes according to the presence of the UBQ domain (Class II), only the HECT domain or three more ARM domains (Class I), the IQ domain (Class III), the UBA and UIM domains, and UBA is always distributed upstream of UIM (Class IV).

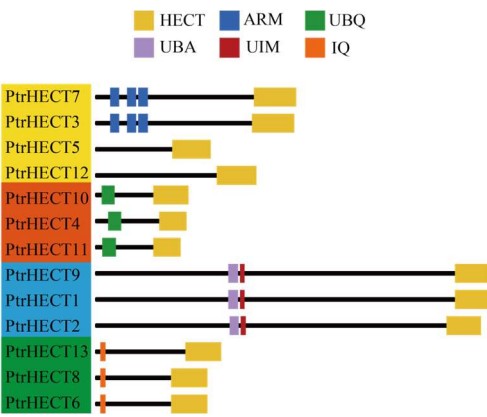

**Figure 3.** The conserved domains of PtrHECT proteins.

### *3.4. Gene Structure and Conserved Motif*

In order to study the correlation between the evolutionary relationship and gene and protein structure, the intron–exon structures and protein-conserved motifs of *PtrHECT* were analyzed (Figure 4). The results showed that the length and number of introns and exons in the same subfamily are similar. For example, the number of exons in subfamily II is three or four, which is significantly lower than the average number in the whole family. All three members of subfamily IV contain an exon with a length greater than 6 kb. It is worth noting that the amino acid length of subfamily IV is much smaller than that of subfamily III; although the gene lengths of subfamilies III and IV are similar, the number of exons of subfamily IV is generally greater than that of subfamily III. This indicates significant differences in the intron–exon structure of *PtrHECT*.

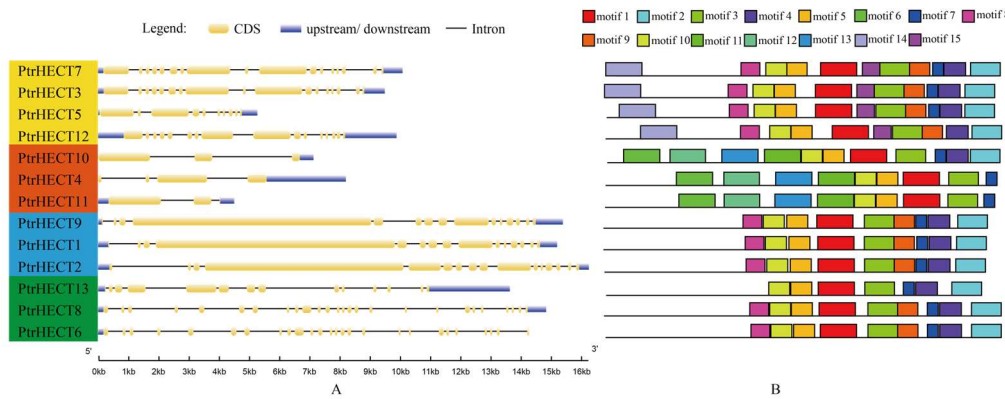

**Figure 4.** Gene structure (**A**) and protein conserved motifs (**B**) of the HECT subfamily in *Populus trichocarpa*.

Conserved motif analysis showed that motif 1, motif 3, motif 5, and motif 10 were common motifs in the 13 members, and motif 1 and motif 3 belonged to the typical HECT domain. Although motif 2, motif 4, and motif 7 do not appear in all members, they are present in all four subfamilies. Motif 14 and motif 15 are unique to subfamily I. Motif 6, motif 11, motif 12, and motif 13 are unique to subfamily II, indicating that there are differences in the types of motifs between subfamilies. The results displayed a difference in motifs between the *PtrHECT* subfamily.

### *3.5. Promoter cis-Element Analysis*

*Cis*-acting elements play an important role in the transcriptional regulation of plants. In order to understand the possible biological functions and regulatory mechanisms of *PtrHECT*, the *cis*-acting elements were analyzed using PlantCARE. The results showed that the promoters of *PtrHECT* mainly included two categories: 12 kinds and 100 *cis*-acting

elements (Figure 5). One is the plant hormone response element, including gibberellin-responsive elements (P-box and TATC-box), methyl jasmonate (MeJA)-responsive elements (CGTCA/TGACG), auxin-responsive elements (TGA), abscisic acid-responsive elements (ABRE), and salicylic acid-responsive elements (TCA). The other is an abiotic stress-related *cis*-element, including hypoxia-inducible elements (GC and ARE), MYB drought-inducible binding sites (MBS), low-temperature-responsive elements (LTR), and defense- and stress-responsive elements (TC-rich repeats).

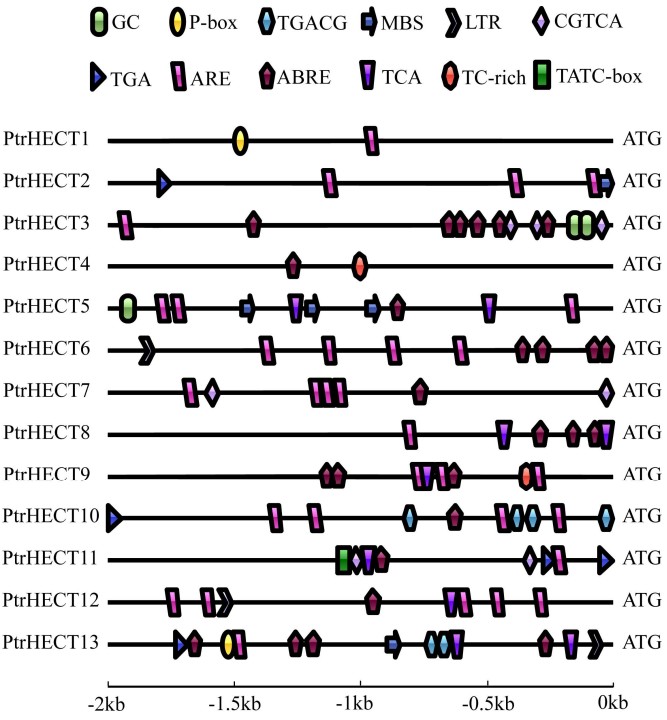

**Figure 5.** Distribution of *cis*-elements in *PtrHECT* promoter regions.

The number of *cis*-acting elements of different types varies greatly. ARE has the largest number of *cis*-acting elements, and 29 of the 13 promoters are distributed among all member promoters except *PtrHECT4*. ABREs have the second largest number, with a total of 26, and the *PtrHECT3* promoter contains the most ABRE, with a total of 6.

### 3.6. Analysis of the HECT Protein Interaction Network in P. trichocarpa

The various life activities of plants depend on protein interactions. Using STRING to predict the protein interaction network of PtrHECT, 13 protein network interaction maps were obtained (Figure 6). We found that there are different degrees of interaction within the PtrHECT protein family, and most of the members interact with ubiquitin-conjugating enzymes, 26S proteasome subunits, and transporters. In the protein interaction networks, the *PtrHECT* proteins of the same subfamily have similar protein interaction networks. Except for members of subfamily II and PtrHECT13, POPTR_0006s08580 and POPTR_0006s08590 interact with all PtrHECT proteins. NRPD903 (a subunit of RNA polymerase IV) interacts with all members of the PtrHECT subfamily II.

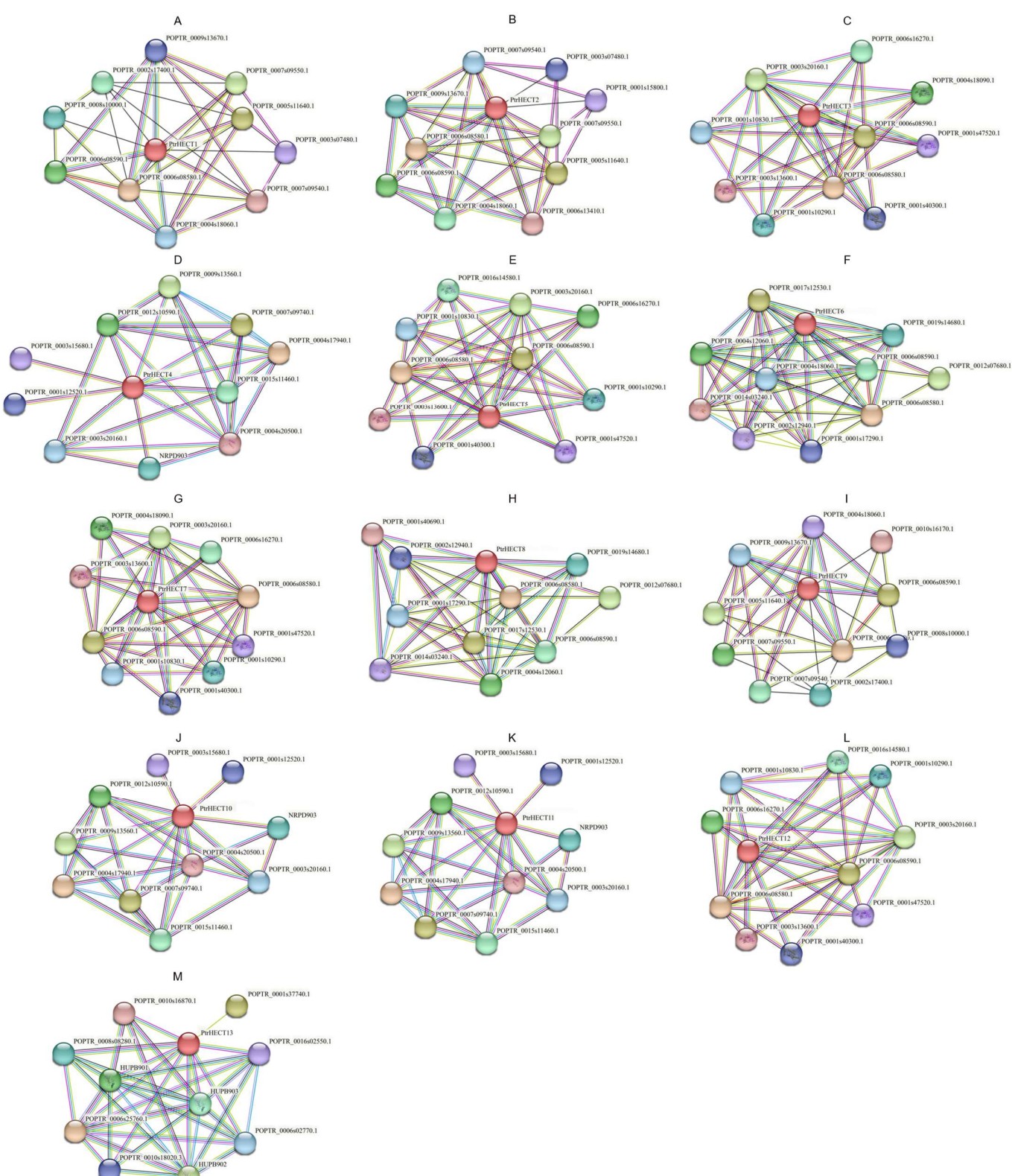

**Figure 6.** The predicted protein interaction networks of PtrHECT proteins (**A–M**).

*3.7. Homology Modeling and Structure Validation of Modeled Proteins*

To understand the structural arrangement and 3D coordination, models of the PtrHECT proteins were generated, respectively (Figure 7). Among them, the 3D structures of PtrHECT6 and PtrHECT8, PtrHECT10, PtrHECT4, and PtrHECT11, which are homol-

ogous proteins, were very similar, and the structures of PtrHECT3 and PtrHECT7 were likewise comparable. PtrHECT5 and PtrHECT12, PtrHECT9, and PtrHECT1 are also homologous proteins, but the similarities between them are poor.

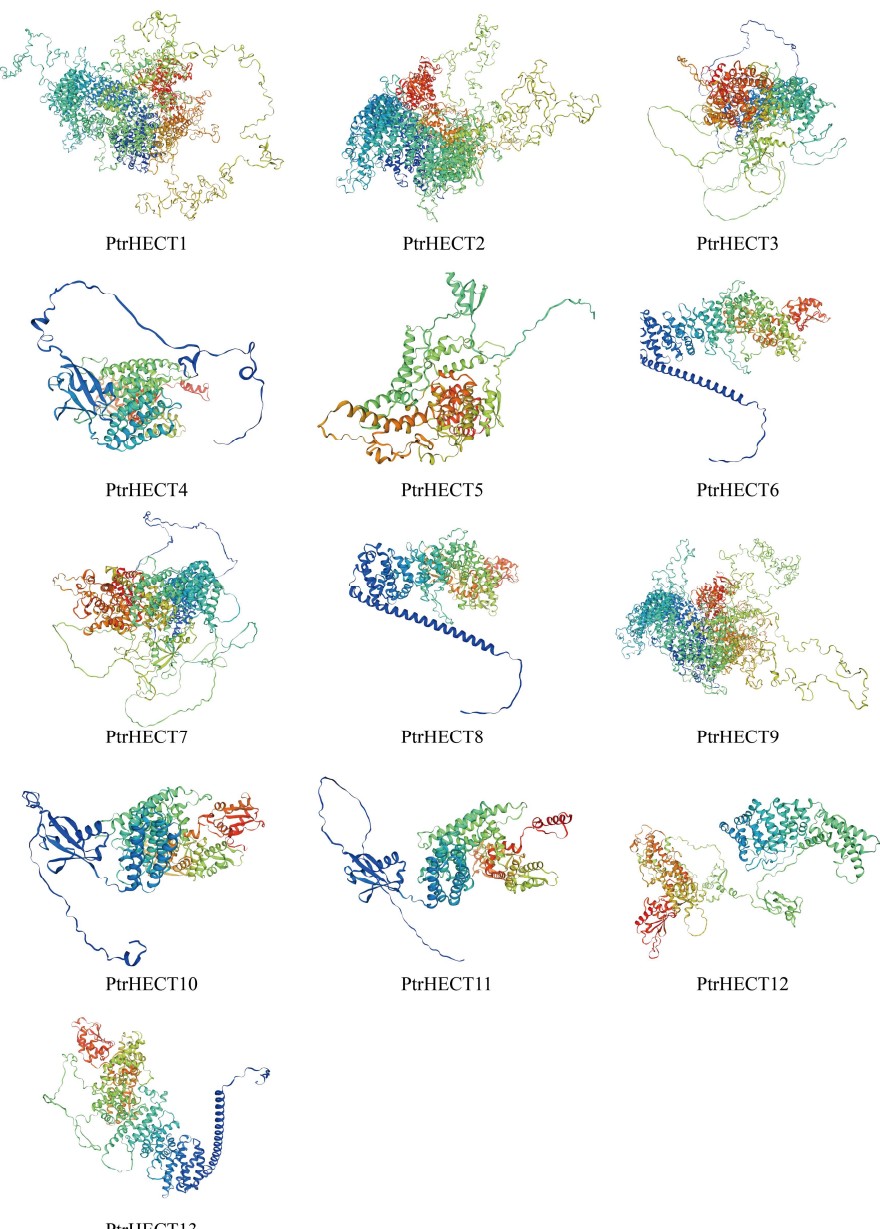

**Figure 7.** Three-dimensional structures of PtrHECT E3 ubiquitin ligase proteins.

The generated homology models were validated using MolProbity Ramachandran plot analysis (Supplementary Figures S2B–S14B), Verify3D (Supplementary Figures S4C–S9C and S11C–S14C), and ERRAT (Supplementary Figures S2C, S3C, and S10C). Both the Ramachandran plot (Supplementary Figures S2A–S14A) and the Ramachandran plot statistics were obtained from the PDBsum web server. The results confirmed the accuracy of the 3D modeling, as the percentages of the residue of the 3D structures modeled by PtrHECT proteins were 78.6%–92.2% in the most favored regions, 6.1% and 16.8% in additional allowable regions, 0.5%–4.3% in generously allowed regions, and 0.3%–4% in disallowed regions, respectively. All these data validate that the modeled 3D structures are essentially high-quality models. The structural variations in the HECT domains may correspond to functional diversity.

### 3.8. Expression Profiles of HECT Genes in P. trichocarpa

We summarized the expression data of *PtrHECT* gene family members in root, shoot, leaf, xylem, phloem, fiber, and vessel in the GEO genome database, set up three replicates, conducted the statistical analysis, and displayed the results by drawing a heatmap (Figure 8). There were significant differences in the expressions of *PtrHECT* genes in the above parts.

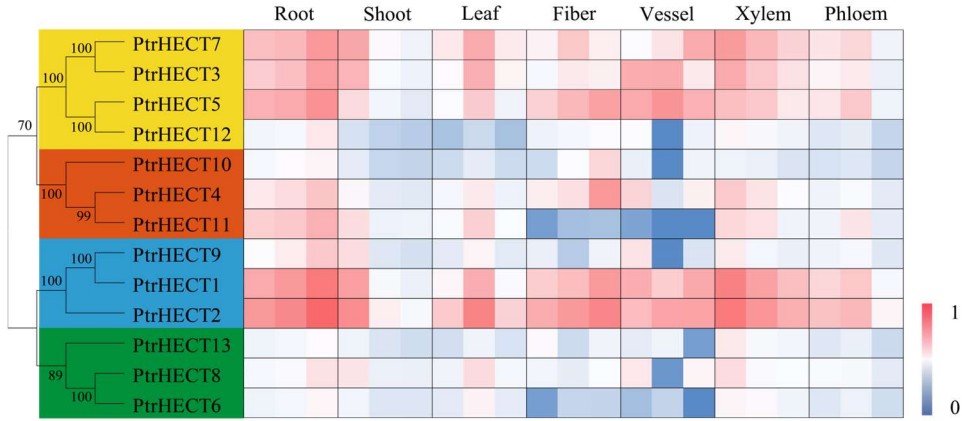

**Figure 8.** Heatmap of *PtrHECT* gene expression patterns in different tissues.

The entire *PtrHECTs* can be divided into two different expression patterns: one is made up of *PtrHECT6* and *PtrHECT11*, whose relative expression levels in vessels and fibers are significantly lower compared to other parts; the remaining genes belong to another category, with little difference in expression between the different parts. Among them, *PtrHECT1* and *PtrHECT2* had the highest expression in roots and xylem, followed by fibers and vessels, and other parts had different degrees of high expression. The relative expression levels of *PtHECT3*, *PtHECT7*, *PtHECT4,* and *PtrHECT5* were weaker than those of *PtrHECT1* and *PtrHECT2*. The remaining genes also exhibited similar expression patterns and had the lowest expression levels in vessels, although these were not statistically significant. In summary, *PtrHECT1*, *PtrHECT2*, *PtrHECT3*, *PtrHECT7*, and *PtrHECT5* had relatively high expression levels in roots and xylem. For almost all genes, the most highly expressed part was the root.

### 3.9. Expression Pattern of PtrHECT Genes under Drought Stress

The results of the expression pattern of *PtrHECT* genes under drought stress are as follows (Figure 9). In roots, *PtrHECT2*, *5*, *6*, *7*, *9*, *11*, and *13* were down-regulated at all time points ($p < 0.01$), while *PtrHECT3*, *4*, *8*, and *12* were rapidly induced 3 h after drought stress and then gradually declined. The expression of *PtrHECT4* was up-regulated to a nearly six-fold level. There was no significant change in the expression level of *PtrHECT1* compared to the control. The expression level of *PtrHECT12* returned to a normal level at 12 h and at later time points, while the expression levels of the remaining genes in the family were still significantly poor at 24 h. In leaves, all genes were induced under drought stress. *PtrHECT1*, *4*, *7*, *8*, and *9* were significantly up-regulated at each time point. The expression of *PtrHECT10*, *11*, and *12* peaked at 6 h, and the expression of *PtrHECT12* was about 60-fold. The expression of all other genes peaked at 24 h, and the expression of *PtrHECT4* was up-regulated 30-fold. *PtrHECT2*, *3*, *5*, and *6* showed no significantly early expression (3 h and 6 h) but peaked at 24 h, suggesting a late response.

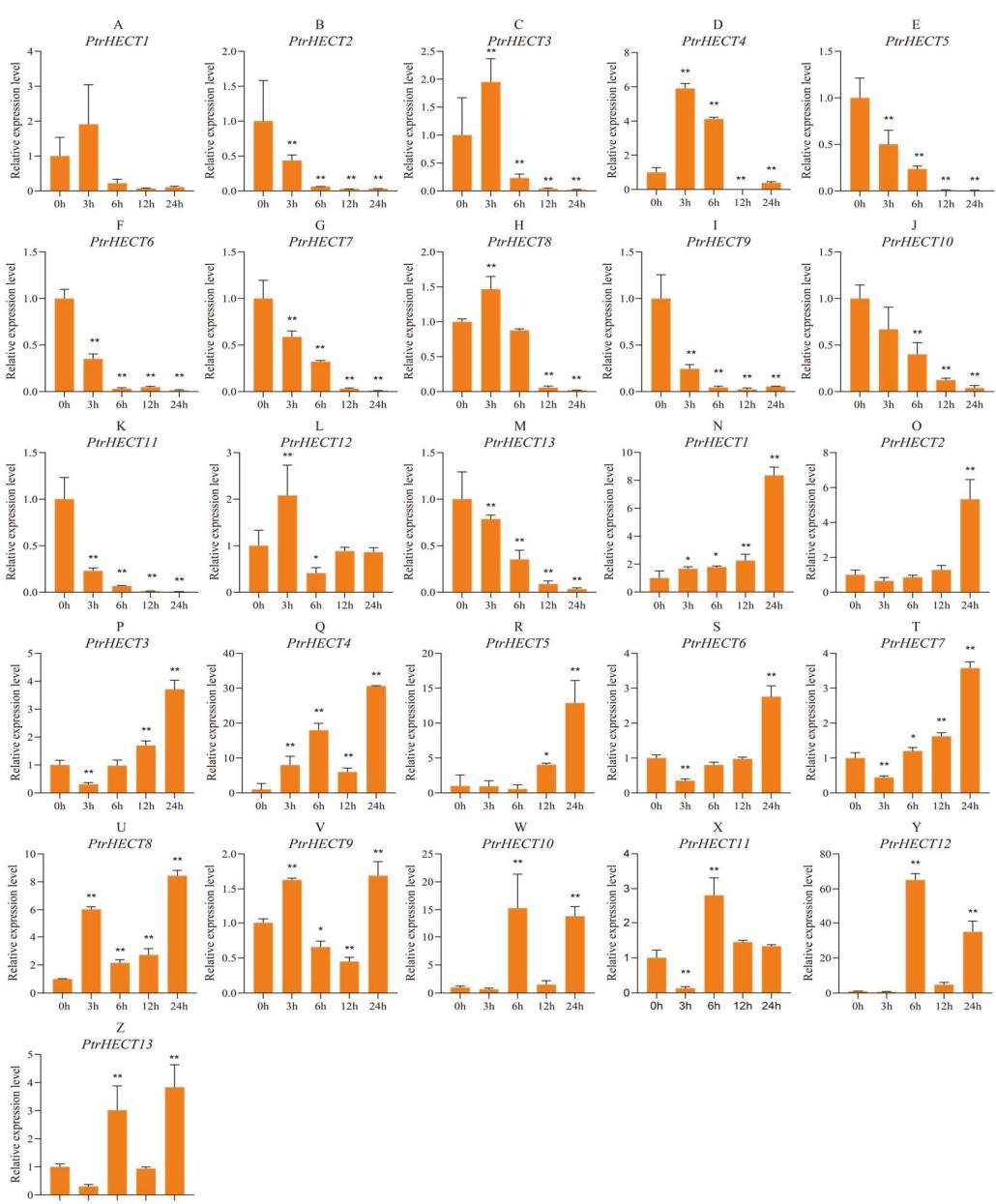

**Figure 9.** The relative expression level of *PtrHECT* genes under drought treatment in roots (**A–M**) and leaves (**N–Z**). Error bars represent deviations from three biological replicates. The *x*-axis represents the stress duration, and the *y*-axis represents the relative expression of the gene. The asterisk represents the difference in transcript abundance from the control (* $p < 0.05$, ** $p < 0.01$).

These results revealed that the expression of most members of the *PtrHECT* family in roots and leaves showed the opposite trend with drought stress, namely that they were down-regulated in roots and up-regulated in leaves.

### 3.10. Expression Pattern of PtrHECT Genes under Salinity

We analyzed the specific expression of 13 genes in roots and leaves under high salt stress (Figure 10). In roots, all members of *PtrHECT* can be induced. The expression of *PtrHECT1*, *2*, *4*, *5*, *7*, *9*, *10*, and *13* was significantly up-regulated at the initial time point (3 h), while *PtrHECT8* was up-regulated at 12 h. *PtrHECT1*, *2*, *3*, *5*, *9*, *10*, *11*, and *13* peaked at 6 h, whereas *PtrHECT6*, *7*, *8*, and *12* peaked at 12 h. In leaves, 10/13 genes were induced. *PtrHECT1* and *2* showed no significant change in expression compared with the control, and only *PtrHECT11* was suppressed. The expression levels of *PtrHECT3*, *4*, *6*, *7*, *8*, *9*, *12*,

and *13* remarkably peaked at 3 h, and then they gradually declined to normal levels. The expression levels of *PtrHECT9* and *12* at 3 h were seven-fold higher compared to thecontrol. The expression levels of *PtrHECT5* and *10* peaked at 6 h and then declined to normal levels.

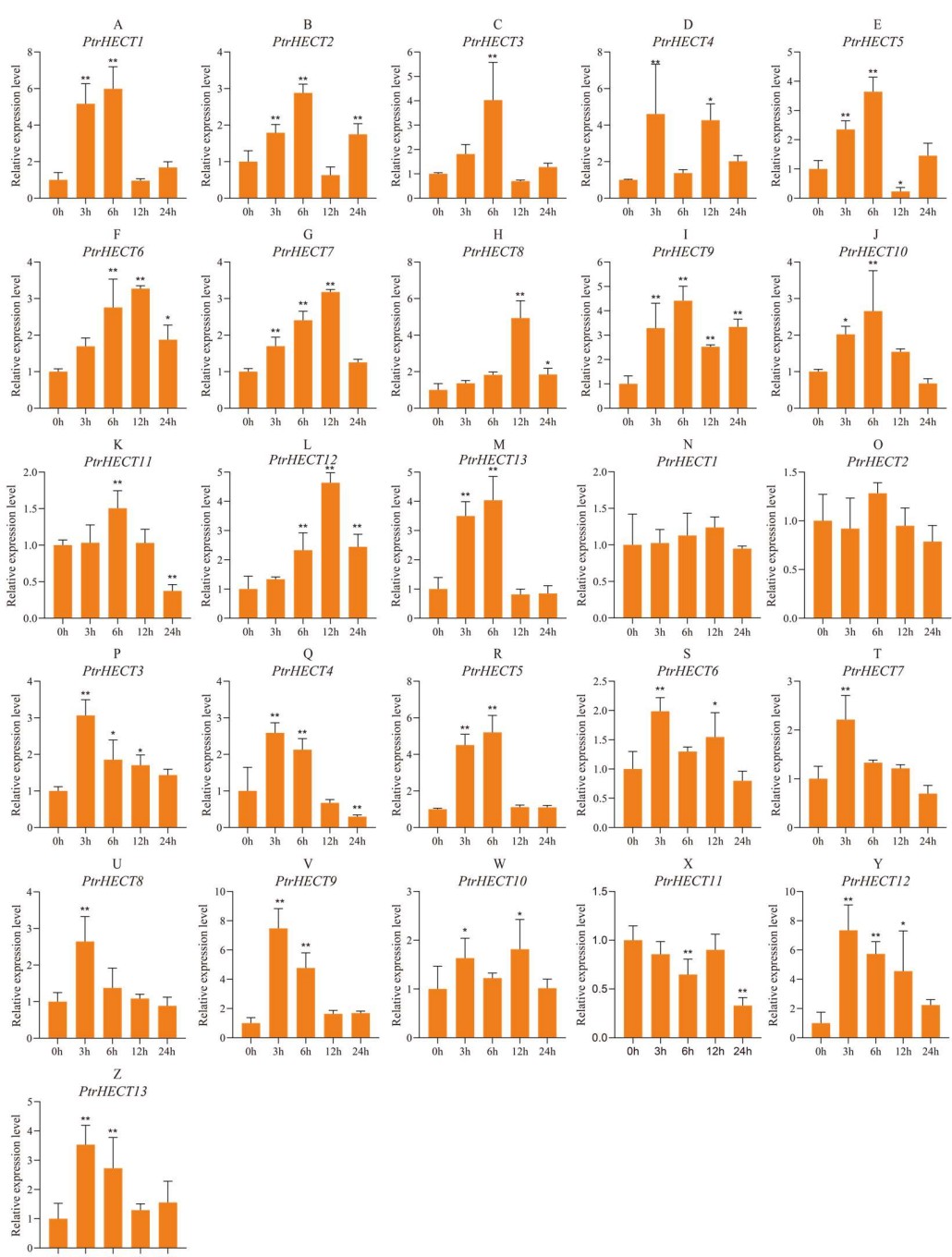

**Figure 10.** The relative expression level of *PtrHECT* genes under salt treatment in roots (**A–M**) and leaves (**N–Z**). Error bars represent deviations from three biological replicates. The *x*-axis represents the stress duration, and the *y*-axis represents the relative expression of the gene. The asterisk represents the difference in transcript abundance from the control (* $p < 0.05$, ** $p < 0.01$).

In summary, six genes, both in roots and leaves, were significantly up-regulated at 3 h (*PtrHECT4*, *5*, *7*, *9*, *10*, and *13*). Almost all the genes in roots peaked at 6 h and 12 h (except for *PtrHECT4*), whereas most of the genes in leaves peaked at 3 h (*PtrHECT3*, *4*, *6*, *7*, *8*, *9*, *12*, and *13*). The results indicate that the *PtrHECT* genes in roots and leaves can respond

quickly to the stress of a high-salt environment, but the up-regulation response in roots lasts longer.

## 4. Discussion

### 4.1. The HECT Genes Family in Populus Trichocarpa

As an important class of E3 ligase in the plant ubiquitin–proteasome, HECT-type ubiquitin ligase has a specific HECT domain, participates in many cellular processes and biochemical reactions, and regulates plant growth, development, and stress response [19]. *HECT* genes exist in the form of a multi-gene family in *Arabidopsis*, tomato, rice, soybean, apple, maize, and other plants [5,21,27,50,51]. However, systematic identification and bioinformatics research of *HECT* genes in the model plant *P. trichocarpa* have not been reported.

In this study, 13 genes were identified in *P. trichocarpa*. The PtrHECT proteins all have a typical HECT domain at the C-terminus and a cysteine-active site. The physicochemical properties revealed that the HECT E3 ligases of *P. trichocarpa* were generally alkaline proteins with low hydrophilicity. Most of the HECT proteins are located in the nucleus and cytoplasm, and a few members are located in the chloroplast, even if the subcellular localizations of homologous genes are different, indicating that they may differ in functions and signal transduction [25]. Chromosomal localization showed that the *HECT* genes were unevenly distributed on the chromosome.

Phylogenetic tree analysis showed that the thirteen members of the *PtrHECT* gene family were divided into four subfamilies. The number of *HECT* genes in soybeans was 1.46 times that in *P. trichocarpa* because segmental duplication contributed significantly to the expansion of the soybean *HECT* gene family. All the *PtrHECT* genes had orthologous genes in soybean, but the homologous gene of *PtrHECT2* in *A. thaliana* (UPL8 in their study) did not exist. This is consistent with the results of previous studies [25]. In previous studies, the *HECT* gene family has been divided into seven subfamilies. The reason for this difference is that our study did not divide the subfamilies I, II, and IV into two groups, as the differences in their conserved motifs and N-terminal structures were not considerable enough to be divided.

Exon–intron structures and protein-conserved motif analysis showed that there were differences and complexity among and within the *PtrHECTs*. In the PtrHECT protein family, we found that there are additional functional domains, and the additional domains of the four subfamilies are inconsistent. PtrHECT3 and PtrHECT7 have three ARMs. Studies have shown that ARM participates in many abiotic stress responses [18]. The ubiquitin-binding regions UBA and UIM are unique to the members of subfamily IV, which are closely arranged in the primary structure of the protein, and UBA is always distributed upstream of UIM, which is similar to UPL1 and UPL2 in *A. thaliana* [20]. The unique UBQ domain of subfamily II is similar to the ubiquitin/ubiquitin-like structure, and the molecular weights of the proteins of subfamily II are much smaller than other PtrHECT proteins, which is similar to the small molecular weight of the ubiquitin protein. This suggests that the *HECT* family of *P. trichocarpa* could potentially encompass a diverse range of functions, and it is postulated that similar gene structures and domains in the same subfamily of *PtrHECT* may interact with the same or similar substrates, causing functional differentiation between subfamilies to a certain extent.

### 4.2. The Promoter cis-Elements of PtrHECT

*Cis*-acting elements regulate the transcription level of genes and enhance plant resistance. We identified at least two phytohormone or abiotic stress-responsive *cis*-elements in all *PtrHECT* family members, indicating that *PtrHECT* genes may play an important role in response to abiotic stress and plant hormone treatment. Salicylic acid (SA) is involved in several physiological processes, including seed germination, growth regulation, flower induction, and particularly in regulating plant responses under stress conditions [52,53]. At present, it has been reported that UPL1, UPL3, UPL4, and UPL5 in *A. thaliana* are involved in the response to SA [6,18]. According to phylogenetic analysis, PtrHECT9 and PtrHECT11

are homologous proteins to UPL1 and UPL5 in *A. thaliana*, respectively, and PtrHECT5 and PtrHECT12 belong to the same subfamily as UPL3. And the *cis*-acting elements of the promoters of *PtrHECT5, 9, 11*, and *12* all contain TCA. This suggests that HECT proteins may serve as SA-responsive elements and participate in regulatory signaling responses. It is particularly important that, according to the results of previous studies, UPL3 in *A. thaliana* can act as a genome-wide amplifier of SA-responsive transcriptional reprogramming and the establishment of immunity, and the expression of downstream genes has undergone profound changes [18]. However, no TCA element was found in the homologous genes of UPL3. Further research is needed to determine whether the absence of TCA elements is involved in the SA regulatory network. Undeniably, HECT-type E3s play an important role in plant resistance to adversity.

### 4.3. Transcript Profiles of HECT Genes under Drought Stress and Salinity

Water plays a decisive role in the survival of plants. A water shortage will affect the osmotic pressure of plants and reduce the absorption and transport of substances, so that normal physiological activities are blocked or even stopped. We analyzed the expression levels of 13 *PtrHECT* genes in roots and leaves under drought and high salt stress using qRT-PCR. Under drought stress, most genes were strongly inhibited in roots but induced in leaves. The inhibition or induction of drought on them was still significant at 24 h, indicating that the damage of drought to plants was persistent and that the response strategies of *PtrHECT* in roots and leaves were different. Under high salt stress, except for *PtrHECT1, 2*, and *11*, other genes were induced significantly in roots and leaves at the time point of stress initiation, though they returned to normal expression levels after a certain period of time, indicating that the *PtrHECT* gene family is generally involved in the regulatory network under high salt stress. Both drought and high salt can lead to an increase in cell osmotic pressure, but the expression of *PtrHECTs* shows a different performance, which indicates that drought and high salt change expression through different regulatory networks.

The root is the first place in which changes in osmotic potential are first sensed, and they quickly transmit signals to other parts [54]. Tissue-specific analysis shows that the root and xylem are the regions with the highest expression of *PtrHECTs*. Thus, it is speculated that under stress, the *PtrHECT* genes with high expression levels in the root respond rapidly. The active HECT proteins in roots regulate the expression of downstream genes by controlling the ubiquitination rates of target proteins and transmitting signals to stems and leaves so that plants can make rapid adjustments to cope with stress, which has a positive effect on plant development and stress resistance.

### 4.4. The Predicted Interaction Network of PtrHECT Proteins

ABA is a key abiotic stress-related hormone involved in various physiological processes, which can promote the senescence and abscission of leaves and fruits [55,56]. During drought, ABA regulates stomatal closure, reduces the rate of dehydration, and improves plant drought resistance. POPTR _ 0006s08580 (PTR6) is a type of E3 ligase that binds to a specific N-degron on the target protein. At present, only two types of PTR proteins have been identified in *A. thaliana*, which are sensitive to ABA and also participate in the physiological process of ethylene-promoted dormancy seed germination [57]. They can regulate seed germination and are widely involved in anti-pathogen, anti-drought, and hypoxic stress-related responses. Considering that multiple members of PtrHECT are predicted to interact with PTR6 and that the *PtrHECT* gene family generally contains the ABRE motif, especially all members of subfamily I, we speculate that most HECT proteins may be involved in ABA-related biochemical reactions to help plants resist drought stress. It has been confirmed that WRKY53 is a key transcription factor that promotes leaf senescence, and UPL5 is an interacting factor of WRKY53 in *A. thaliana* [58]. UPL3 and UPL4 are also involved in leaf growth and senescence regulation. It can be speculated that the subfamily I proteins of *P. trichocarpa*, which are clustered with UPL3 and UPL4, can participate in regulation through the ABA pathway. In addition, we predicted that the members of the

subfamily II of PtrHECT proteins all interact with NRPD903, which has been identified as a subunit of RNA polymerase IV [31]. Moreover, PtrHECT4 and 11 in subfamily II have a unique UBQ structure. Further evidence is needed to determine whether the above factors collectively contribute to changes in the functions of PtrHECT4 and 11.

## 5. Conclusions

Herein, 13 identified *PtrHECT* genes were classified into four phylogenetic groups. The analysis of the conserved structure revealed structural differences between the *PtrHECT* gene subfamilies. The *cis*-acting elements and protein–protein interaction network revealed the response of *PtrHECT* genes to ABA and SA. To the best of our knowledge, this is the first time that *HECT* genes in *P. trichocarpa* have been analyzed in light of responses to salt and drought based on qRT-PCR analysis, and the majority of *PtrHECT* genes were responsive to drought and high salt in leaves. Thus, our study provides a theoretical basis for the future study of *HECT* genes involved in the signaling pathways of ABA and SA under stress and the determination of the specific function of each *PtrHECT*, which is helpful for the selection of suitable candidate genes for the genetic engineering of drought and high salinity resistance in *P. trichocarpa*.

**Supplementary Materials:** The following supporting information can be downloaded at: https://www.mdpi.com/article/10.3390/f15010169/s1, Table S1: The primers for qRT-PCR; Figure S1: Sequence alignment of the HECT domain of all the identified HECT proteins in *Populus trichocarpa*, *Arabidopsis thaliana*, *Glycine max*, *Oryza sativa*, *Zea mays* and *Triticum aestivum*; Figure S2: Homology modelled structure validation of PtrHECT1 using (A) Ramachandran plot; (B) Ramachandran plot statistics; and (C) ERRAT; Figure S3: Homology modelled structure validation of PtrHECT2 using (A) Ramachandran plot; (B) Ramachandran plot statistics; and (C) ERRAT; Figure S4: Homology modelled structure validation of PtrHECT3 using (A) Ramachandran plot; (B) Ramachandran plot statistics; and (C) Verify3D; Figure S5: Homology modelled structure validation of PtrHECT4 using (A) Ramachandran plot; (B) Ramachandran plot statistics; and (C) Verify3D; Figure S6: Homology modelled structure validation of PtrHECT5 using (A) Ramachandran plot; (B) Ramachandran plot statistics; and (C) Verify3D; Figure S7: Homology modelled structure validation of PtrHECT6 using (A) Ramachandran plot; (B) Ramachandran plot statistics; and (C) Verify3D; Figure S8: Homology modelled structure validation of PtrHECT7 using (A) Ramachandran plot; (B) Ramachandran plot statistics; and (C) Verify3D; Figure S9: Homology modelled structure validation of PtrHECT8 using (A) Ramachandran plot; (B) Ramachandran plot statistics; and (C) Verify3D; Figure S10: Homology modelled structure validation of PtrHECT9 using (A) Ramachandran plot; (B) Ramachandran plot statistics; and (C) ERRAT; Figure S11: Homology modelled structure validation of PtrHECT10 using (A) Ramachandran plot; (B) Ramachandran plot statistics; and (C) Verify3D; Figure S12: Homology modelled structure validation of PtrHECT11 using (A) Ramachandran plot; (B) Ramachandran plot statistics; and (C) Verify3D; Figure S13: Homology modelled structure validation of PtrHECT12 using (A) Ramachandran plot; (B) Ramachandran plot statistics; and (C) Verify3D; Figure S14: Homology modelled structure validation of PtrHECT13 using (A) Ramachandran plot; (B) Ramachandran plot statistics; and (C) Verify3D.

**Author Contributions:** Conceptualization, Y.F. and J.Y.; methodology, H.L. and Y.L.; software, H.Z.; validation, D.Y. and H.Z.; formal analysis, H.L.; investigation, Y.F.; resources, Y.L.; data curation, Y.L.; writing—original draft preparation, Y.F.; writing—review and editing, Y.F.; visualization, H.L.; supervision, H.Z.; project administration, D.Y.; funding acquisition, A.C. and J.Y. All authors have read and agreed to the published version of the manuscript.

**Funding:** This work was supported by the National Natural Science Foundation of China (No. 31870649) and the Innovation Project of the State Key Laboratory of Tree Genetics and Breeding (Northeast Forestry University) (2021A02).

**Data Availability Statement:** The research data are available upon request from the corresponding author.

**Conflicts of Interest:** The authors declare no conflicts of interest.

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
