# Peer review of "Genome-Wide Analysis of Homologous E6-AP Carboxyl-Terminal E3 Ubiquitin Ligase Gene Family in Populus trichocarpa"

_forests, doi:10.3390/f15010169_

Round 1

Reviewer 1 Report (Previous Reviewer 1)

Comments and Suggestions for Authors

The abstract should be improved. This is very insufficient in particular: "Under salt stress, all members of PtrHECT can be induced in roots." This is a very vague and general sentence. Also, you repeat this same sentence in the results. Avoid doing this. Please, re-phrase this sentence and improve the entire abstract. The abstract serves to draw the reader's attention. This abstract in its present form doesn't do that. 

L102 - use Latin names for all taxons, e.g., corn (Zea mays), wheat (Triticum aestivum

L330-334 it would be good to name these genes. This is the Results section, you should be as precise about presenting your results as possible.

L360 can you speculate on why P. trichocarpa HECT protein evolved in this way? In general, I think you should discuss the evolution of these proteins in P. trichocarpa more.

L412-413 sentence very unclear due to less than perfect English.

Line 440 this is unclear. HECT genes are ubiquitous. It is not surprising that P. trichocarpa also has them. Re-phrase this sentence, e.g., "to the best of our knowledge, this is the first time that HECT genes in P. trichocarpa are analyzed in light of responses to salt and drought". 

Line 442 what useful information? This should be written here, concisely.

Line 443 re-write this sentence so that it contains a real conclusion based on your research. You said that based on cis-elements, you found that at least two of these genes are responsive to salicylic acid (you said this in L378-379). This is important. Highlight it here. 

Line 445 specify how. How can P. trichocarpa be used for these studies? Based on which observations from your study?

Comments on the Quality of English Language

I suggest again that the manuscript be edited by a native-level English speaker. The manuscript was significantly improved since the last version, but there is room for improvement. In some sections, the manuscript is relatively hard to follow due to flawed English.

Author Response

Dear reviewer: Thank you for the positive assessment of our work, and your suggestions on how it can be improved. Due to your suggestion, we have re-phrased sentences and improved the entire introduction, abstract and conclusion, and corrected the mistakes in the manuscript.

Reviewer 2 Report (Previous Reviewer 2)

Comments and Suggestions for Authors

The authors addressed all the comments properly and the revised manuscript is suitable for publications. 

Comments on the Quality of English Language

NIl

Author Response

Dear reviewer: Thank you for the positive assessment of our work, and your suggestions on how it can be improved in the past.

Reviewer 3 Report (New Reviewer)

Comments and Suggestions for Authors

Fu et al. identified 13 HECT E3 ubiquitin ligase genes in the Populus trichocarpa genome. Phylogenetic analysis showed that these genes were divided into four subfamilies. Poplar plants in vitro demonstrated different expression patterns of PtrHECT genes under drought and sat stresses. There are several minor comments on the manuscript.

The paragraph listing plant species with identified HECT genes (L. 56-66) could be reduced.

L. 67. “Popolus” should be corrected to “Populus”.

L. 147. A reference to Woody Plant Medium must be added.

L. 148. “in vitro” should be italicized.

L. 152. “leave” should be corrected to “leaves”.

Fig. 2. Why were HECT genes from a woody plant, such as apple, not used to construct a phylogenetic tree?

Author Response

Dear reviewer: Thank you for your insightful comments and suggestions on our manuscript. According to your suggestions, we have added the HECT gene in apple to the construction of the evolutionary tree and corrected the mistakes in the manuscript.

Round 2

Reviewer 1 Report (Previous Reviewer 1)

Comments and Suggestions for Authors

Dear Authors,

I see you made significant improvements since Round 2. In my opinion, the manuscript can be published now after minor English language editing. 

Comments on the Quality of English Language

Minor English language editing is required.

Author Response

Dear reviewer: Thank you for the positive assessment of our work and your insightful suggestions on how it can be improved in the past. The manuscript has undergone English language editing by MDPI and then we conducted minor English language editing on it (revisions to the manuscript have been highlighted).

This manuscript is a resubmission of an earlier submission. The following is a list of the peer review reports and author responses from that submission.

Round 1

Reviewer 1 Report

Comments and Suggestions for Authors

The manuscript "Identification and tissue-specific expression analysis of PtrHECT 2 gene family in Populus trichocarpa" is an informative pilot study of genes of interest responsible for stress responses in the plant P. trichocarpa. The research design and methodology, particularly bioinformatics, are mostly appropriate. However, the authors did not present statistical analysis for gene expression and this is the basis for Major Revision. This must be corrected. Minor revisions: names of taxons must be written in Italics (the lack of Italics throughout the manuscript should be corrected). 

Abstract: medium English editing is needed. Also, since there's no statistical analysis of gene expression in this version of manuscript, the Abstract must be also revised.

L61 - Brassica in Italic

L94 - P. trichocarpa in Italic 

L102 - since it's a taxon's name in Latin, Oryza sativa must be in Italic

L100 - details about the phylogenetic tree are missing. How were the distances computed? Which method was used for evolutionary distances? 

L124 - regarding Expression Analyses, statistical analysis is missing. Did the authors conduct statistical analysis? It should be done, otherwise, it's impossible to reliably compare the expression of different genes. 

L142 - format the table so that's in line with the rest of the text.

L243 - is it justified to say that the differences in expression were significant if no statistical analysis was presented? Heatmaps are usually accompanied by a graph plotting the p-value of differences between the samples (vertical axis) against the factor of change in expression (y-axis). 

L285 - this is speculated by whom? The authors or are authors citing other sources? Also, I think that this conclusion is too strong based on this relatively small sample and the lack of functional analysis. Genetic variation doesn't necessarily mean different functions. Try to rephrase this. 

L314 - in line with the previous comment, please revise the Discussion so that it includes statistical analysis. 

L323 - please revise the Conclusions to reflect statistical analysis.

Comments on the Quality of English Language

English needs medium editing. 

Reviewer 2 Report

Comments and Suggestions for Authors

General Comments: In the manuscript, authors retrieved 13 HECT genes from the genome of Populus trichocarpa and analyzed their features using insilico methods. The present manuscript contains very basic bioinformatics data. For example, prediction of cis-acting elements, mapping of chromosomal location, identification of subcellular localization, intron and exon region and more are very basic bioinformatics data. Authors should validate the HECT genes using qRT-PCR or digital droplet PCR in populus under various abiotic stress conditions or at least one specific abiotic stress condition (drought or salinity). Otherwise the present manuscript is not suitable for publication in the journal of forests. Apart from this, authors should recreate the phylogentic tress with more plants. Authors should identify what are the functional residues are commonly present in the HECT genes of populus and other plants through clustal alignment. Authors should develop homology modelling for HECT genes. So, overall the manuscript does not contain any new information and not ready for publications. I recommend to the authors to conduct laboratory experiment and analyzed the expression of pattern of HECT genes under abiotic stresses. Also, authors should develop physiological, biochemical and morphological traits for this experiment. The manuscript contains several minor mistakes some are listed below.

Specific Comments

1.       Authors should mention the full name of HECT in L. No. 10

2.       L. No. 12; based on the phytozome database or from the phytozome database or from genome sequences of Populus trichocarpa at pytozome database? Authors should check and fix the error.

3.       L. No. 14-15, authors should mention the software names in the appropriate analysis. For example, subcellular localization (WoLFPSORT)

4.       L. Nos. 15-17; authors should rewrite the sentences

5.       In the abstract section, authors should mention the expression pattern of HECT genes.

6.       Write the keywords in alphabetical order

7.       Write the full form of WRKY in L. No. 58

8.       Remove extra comma in L. No. 65

9.       Write the common names of glycine max and other plant. Because authors provided both common and botanical name of whet, apple and more.

10.   L. No. 67, through expression analysis in apple or any other plant? Kindly clarify the sentence

11.   L. Nos. 80 and 81, rectify the parenthesis error. The same error occurred in many places. Kindly check and rectify all the errors throughout the manuscript.

12.   L. N0. 94; botanical name of poplus should be in italics and authors should write P. trichocarpa or poplus

13.   How did authors identify 13 HECT genes using 8 HECT sequences of Arabidopsis? Authors should clarify this.

14.   L. No. 101. Authors should mention the botanical name of Arabidopsis where it is used first.

15.   L. No. 102; botanical name rice should be in italics

16.   L. No. 103; what did the authors mean verification parameter?

17.   Why authors did not descript HECT genes of rice in the introduction section. Authors should include in the appropriate place.

18.   What did authors mean motif in section 2.4 and cis-acting elements in section 2.5?

Comments on the Quality of English Language

Authors should check the entire manuscript and rectify all the errors. 

Round 2

Reviewer 1 Report

Comments and Suggestions for Authors

General comments: authors should again revise the nomenclature of genes and proteins. I am not sure if it is consistent throughout the text. In the title: "tissue-specific". Add the hyphen.

Abstract - The abstract shouldn't be a digest of methodology and results, but a highlight of the importance of the study. Authors mostly repeat their methods and results with little emphasis on the scientific importance of their research. Instead of simply repeating their results, the authors should try and rewrite the abstract so that it emphasizes the scientific contribution of their findings. Line 19 - please, rephrase this sentence without a colon. 

Introduction

Line 77 - I would recommend that authors rephrase this sentence and not end it with etc. Either precisely name all gene elements that you analyzed or name it more broadly, e.g. "We analyzed gene elements responsible for the expression and regulation of HECT protein domains". You're introducing the focus of your study here, it shouldn't just be ended with an etc. 

Results

Figure 3 - can the image quality be improved? The legend accompanying motifs has a very small font size, it's hard to read.

Discussion

Perhaps the discussion can be divided into subsections since it's discussing multiple different areas.

Comments on the Quality of English Language

English language editing is needed, preferably by a native-level speaker. 

Reviewer 2 Report

Comments and Suggestions for Authors

I have checked the author's response. Authors did not respond properly. I have suggested to the authors do laboratory experiments but they have skipped all my major comments. They responded with only minor comments. So, I strongly recommend that kindly reject the manuscript without any further chances. How did the authors skip all my major comments? Furthermore, they did not respond to all my comments properly.  I spent my whole day reviewing the manuscript but they have simply skipped all my major comments. This is the worst behavior in research. The current manuscript does not contain any new information and is not suitable for publication in the journal of Forests. 

  Comments on the Quality of English Language

Should be improved